# Molecular Weight Distribution of Humic Acids Isolated from Calcic Cryosol in Central Yakutia, Russia

**DOI:** 10.3390/molecules29133008

**Published:** 2024-06-25

**Authors:** Vyacheslav Polyakov, Evgeny Abakumov, Evgeny Lodygin, Roman Vasilevich, Alexey Petrov

**Affiliations:** 1Department of Applied Ecology, Faculty of Biology, St. Petersburg State University, 7/9 Universitetskaya emb., 199034 St. Petersburg, Russia; e.abakumov@spbu.ru; 2Arctic and Antarctic Research Institute, Beringa 38, 199397 St. Petersburg, Russia; 3Institute of Biology, Komi Science Center, Ural Branch, Russian Academy of Sciences, 28, Kommunisticheskaya St., 167982 Syktyvkar, Russia; lodigin@ib.komisc.ru (E.L.); vasilevich.r.s@ib.komisc.ru (R.V.); 4Research Institute of Applied Ecology of the North, Professor D. D. Savvinov SVFU, Lenin Ave. 43, 677027 Yakutsk, Russia; petrov_alexey@mail.ru

**Keywords:** fallow soil, cryogenic soil, cryolithozone, arable lands, Cryosols

## Abstract

The transition of soils into fallow state has a significant impact on the accumulation and transformation of soil organic matter (SOM). However, the issue of SOM transformation as a result of soil transition to fallow state in cryolithozone conditions is insufficiently studied. The aim of this study is to investigate the molecular weight (MW) distribution of humic acids (HAs) isolated from soils of central Yakutia. Native, fallow and agricultural soils in the vicinity of Yakutsk city were studied. MW distributions of HA preparations were obtained on an AKTAbasic 10 UPS chromatographic system (Amersam Biosciences, Uppsala, Sweden) using a SuperdexTM 200 10/300 GL column (with cross-linked dextran gel, fractionation range for globular proteins 10–600 kDa). The data on the molecular-mass distribution of HAs of fallow and agricultural soils of Central Yakutia were obtained for the first time. According to the obtained data, it was found that the highest carbon content in the structure of HAs was observed in agricultural soils (52.56%), and is associated with soil cultivation and fertilizer application. Among the HAs of fallow soils, we note that those soils that are in the process of self-vegetation have a relatively high carbon content in the HAs (45.84%), but the highest content was observed in fallow soils used as hayfields (49.98%), indicating that the reinvolvement of agriculture in fallow soils leads to an increase in the carbon content of HAs. According to the data of the MW distribution of HAs, it was found that the highest content of a high MW fraction of HAs was recorded in native soil (18.8%); this is due to the early stages of humification and the low maturity of organic matter. The highest content of a low MW fraction of HAs was recorded in agricultural soil (73.3%); this is due to the formation of molecular complexes of a “secondary” nature, which are more stable in the environment than the primary transformation products of humification precursors. The molecular composition of the HAs of fallow soils in the process of self-overgrowing is characterized by values closer to the HAs of native soils, which indicates their transformation towards HAs of native soils. The obtained results indicate that the reinvolvement of fallow soils leads to the transformation of the molecular composition of HAs towards HAs of agricultural soils, and to an increase in the resistance of SOM to biodegradation.

## 1. Introduction

Soil organic matter (SOM) and its component humic acids (HAs), is one of the most important factors determining soil condition, such as fertility, buffer capacity, water-holding capacity, and also plays an essential role in the regulation of the climate on the planet [1,2]. A large number of works have been aimed at studying the composition and quality of the SOM of native and anthropogenically transformed soils [3,4,5]. However, the issues of the molecular structure of HAs and SOM formed in fallow soils in the cryolithozone remain insufficiently studied [2,6], especially the quantitative and qualitative composition of carbon [7,8]. Investigation of the molecular composition and structure of SOM and HAs provides insight into the levels of organic matter transformation, which can significantly improve our understanding of the greenhouse gas potential of permafrost-affected soils [9,10]. Earlier studies show that an increase in the content of the low MW fraction indicates an increase in the proportion of cyclic structures as well as carboxyl groups in the composition of HAs, which is due to the increasing age and stabilization of organic matter [4]. Agriculture has a significant impact on the molecular structure of SOM and HAs; the application of organic and inorganic forms of fertilizers leads to an increase in the pool of soil carbon as well as aromatic structural fragments, which is due to the stabilization of carbon in the soil [11]. However, the role of different structural fragments, with different MWs, in the process of carbon uptake has not been sufficiently studied. A number of researchers have studied the effect of fertilizers on the qualitative composition of HAs and how their transformation occurs depending on the forms of fertilizers [12,13,14], but the mechanism of HA transformation as a result of a change in land use has not been revealed [15], namely, the transition of agricultural soils to fallow state under the influence of cryogenesis. Russia has the largest area of fallow land in the world; this is due to the long history of land development and then its large-scale transition to a fallow state as a result of various socio-economic factors [16]. The problem of soil transition in agricultural land is widespread not only in Russia, but also in Western and Eastern European countries as a result of increasing urban agglomerations, as well as the development of linear infrastructure [17,18]. In the period between 1990 and 2006, up to 122,000 ha of land was removed from agriculture in Europe [19]. The intensification of agriculture in recent decades has led to the problem of agricultural land degradation, which is expressed in erosion, salinization and heavy metal contamination of soils; thus, countries such as the USA and China have had to resort to the practice of forced soil conversion in order to preserve soil fertility [20].

The transition of agricultural soils into a fallow state results in changes in the physicochemical parameters of soils, the qualitative composition of SOM, afforestation, silting, and dehumification [21]. The duration of the soil’s transition to a fallow state plays an important role in the features of SOM accumulation and transformation [6]. In the first 5–15 years, there are no significant changes in the content of SOM, but after 20–40 years, in taiga zone, there is a significant input of organic matter into the soil, which exceeds their losses due to mineralization [22,23]. The reinvolvement of fallow soils into agriculture leads to significant changes in the accumulation and transformation of SOM; this is due to changes in plant composition (precursors of humification), soil type, local climatic features, and ecosystem productivity [24]. A characteristic feature of the cryolithozone is the presence of permafrost, which significantly affects the transformation of fallow ecosystems [2]. The formation of agricultural land is associated with the removal of natural vegetation and ploughing of the territory, in the conditions of the cryolithozone, such activities cause the development of thermokarst processes in the soil that lead to waterlogging of the territory, and the development of frost heaving and fracturing [25]. Thus, as a result of soil cryogenesis development in the soils of fallow ecosystems, a transformation of SOM occurs in soils of fallow ecosystems, which can lead to a decrease in the degree of carbon stabilization in the soils and the emission of climate-active gases into the atmosphere. The analysis of HA fractionation is one of the most effective mechanisms for analyzing the structure of humic substances [26]. By means of HA fractionation, it has been reliably determined that HAs show a polydisperse composition and are able to realize a large number of ecological functions in the soil [4]. The molecular composition of HAs determines such processes as their ability to accumulate and migrate in aquatic environments, their use by microorganisms and plants as a source of nutrients, and the resistance of HAs to biodegradation processes. The method of liquid–gel chromatography for the analysis of the MW fractions of HAs enables complete information on the MW distribution curve of HAs [27] to be obtained. An important issue is to obtain information on the stabilization of organic matter in cryolithozone soils, as it has been found that soils can lose a significant amount of carbon through dehumification during the transition to fallow state [7]. Thus, the aim of this work is to study the MW distribution of HAs to assess the degree of HA stabilization of fallow, native and agricultural soils formed under cryolithozone conditions. For this purpose, the following tasks were set.

-Study the elemental composition of HAs of soils of central Yakutia;-Analyze the MW distribution of HAs;-Evaluate the degree of stabilization of SOM based on the elemental composition and MW distribution of HAs.

## 2. Results and Discussion

### 2.1. Features of Soil Formation in Central Yakutia

The studied soils are represented as fallow (Y1–Y2), reintroduced (Y5–Y6), agricultural (Y7, Y9–Y10) and native soils (Y3–Y4, Y8). The territory adjacent to the city of Yakutsk was subjected to large-scale agricultural development during the Soviet period, and today most of these territories are in the process of self-restoration and self-overgrowth. The soils of fallow lands are characterized by the presence of pronounced old plough horizons from 20 to 30 cm, underlain by undisturbed soil horizons; it is also typical for reclaimed soils, that 30 years later, there are clear signs of agricultural development of these lands. Native soils (Y3–Y4, Y8) are represented by Calcic Cryosol; these soils are characterized by the processes of the accumulation of coarse humus and an eluvial–illuvial distribution of carbonates along the profile. In soils of this type, low-thickness humus horizons up to 15 cm are formed, consisting mainly of poorly decomposed organic residues. Soils subjected to wildfires (Y4) were also used for HA analysis for the purpose of evaluating the influence of high temperatures on HA’s MW distribution. The fallow soils used as pasture (Y5) were characterized by a thick organic-accumulative horizon with signs of over-compaction as a result of cattle grazing. The fallow soil used as a hayfield (Y6) was characterized by a thick organo-accumulative horizon up to 24 cm with a high degree of organic matter transformation. In the lower part of the profile, there were signs of water logging conditions and development of gley process. Soils used in agriculture (Y7, Y9–Y10) are characterized by the formation of thick organo-accumulative horizons up to 55 cm with a dusty and lumpy structure. Soils developing on the first terrace of the river (Y8–10) are characterized by a local accumulation of salts and the formation of salt crust on the soil surface, which negatively affects the development of agricultural crops. Saline soils are predominantly located on floodplain terraces of large river valleys and in depressions of the ancient alluvial plain (alas). The genesis of soil salinization in central Yakutia is related to the history of the formation of the ancient alluvial plain, which is now represented by different-aged areas and terraces of river valleys formed under different bioclimatic conditions. The first floodplain terrace is characterized by the sodium chloride type of salinization [28]. The depth of occurrence of the permafrost ranges from 1.5 to 2 m. 

Transition to fallow state and agricultural development have a significant impact on the transformation of organic matter, which is affecting the molecular composition of HAs and the degree of stabilization of organic matter.

### 2.2. Element Composition of HAs Isolated from Soils of Central Yakutia

The elemental composition of HAs is the most important indicator determining the processes of humification, oxidation and degree of condensation of HAs. The data on the elemental composition of the studied HAs are presented in Table 1.

As a result of the study of the obtained data, it was noted that the lowest carbon content in the studied HAs was observed in soils exposed to wildfire (Y4), the carbon content is 41.62%. The carbon content in HAs of post-pyrogenic soils decreases in relation to native undisturbed soils. According to Dymov et al. [29], the formation of stable aryl fragments was found to occur as a result of fire exposure on SOM. Among the soils not exposed to fire, it can be noted that the highest carbon content is found in the HAs of agricultural soil (Y7, Y9–Y10), which is associated with soil cultivation and fertilizer application. The carbon content in HAs of fallow and agricultural soils is characterized by a relatively high level and ranges from 45.57 to 52.56%. HAs of native soils (Y3, Y8) are characterized by relatively lower carbon content (46.36–49.83%), except for HAs of soil formed on the first terrace of the Lena River; this may be due to the close location of agricultural areas and transfer of organic and inorganic fertilizers to the adjacent area. In terms of nitrogen content in the HAs of soils, we note that the highest content was recorded in the HAs of fallow soils used as hayfields (Y6), indicating high rates of nitrogen fixation from the atmosphere and the formation of nitrogen-containing structural fragments. The lowest content was recorded for pyrogenic native soil (Y4); this is due to the effect of high temperatures on SOM. We calculated the H/C index, which can serve as an indirect index showing the stability of HA molecules to biodegradation; thus, the lower the H/C index, the higher the stability of SOM. According to the data obtained, the least stable HAs are formed in pyrogenic soil; the increase in the degree of hydrogenation is the result of the attachment of hydrogen molecules to carbon and the formation of aliphatic structural fragments in HA composition. Correspondingly, dehydrogenation indicates the detachment of hydrogen molecules and the carbonization of HA molecules, which gives it a more stable form. Thus, the lowest H/C index corresponds to HAs of soils of the experimental vegetable garden (Y7), HAs of agricultural soils are the most stable in the environment. The index w indicates the conditions in which the HAs are located (oxidative/reduction), according to this index, all studied HAs are located in oxidative conditions, which is caused by active rates of humification of SOM.

A van Krevelen diagram was used to graphically represent the elemental composition and the contribution of oxidation/reduction and hydrogenation/dehydrogenation processes in HA molecules (Figure 1).

Based on the obtained data, we can identify a cluster of HAs of agricultural soils (Y7, Y9–Y10) characterized by the process of dehydrogenation and, accordingly, more resistant to biodegradation in relation to the SOM of fallow and native soils. A decrease in the H/C index indicates an increase in the maturity of SOM and the presence of aromatic structural fragments [30]. The change in the elemental composition of HAs is significantly influenced by soil type as well as agricultural land development [31]. Our results are supported by data of Sanchez-Monedero et al. [32], who found that the HAs of agricultural soils are more mature relative to the same soil types that are not subjected to agricultural development. A similar conclusion was obtained by Cui et al. [33], who showed that the HAs of forest ecosystems adjacent to agricultural fields are characterized by the formation of aliphatic structural fragments, while the HAs of agricultural soils are characterized by a large number of new stable components, which determines their high level of maturity. Lodygin and Vasilevich [34] found that arable soils of the taiga zone are characterized by an increase in the content of aromatic structural fragments in relation to natural soils, this is due to the relatively low content of carboxyl functional groups, which contributes to the accumulation of aromatic structural fragments in arable soil horizons. The formation of agricultural soils, associated with the increased biological activity of soils, and leads to an increase in the content of aromatic structural fragments, due to the detachment of peripheral chains in HA molecules, indicating the fact that the agricultural development of soils leads to an increase in the resistance of organic matter to degradation processes [35].

### 2.3. MW Distribution in HAs Isolated from Soils of Central Yakutia

Three regions of molecular fraction distribution were identified on the obtained spectra: high MW region, medium MW and low MW (Figure 2). The obtained MW distributions of HAs were divided into four figures; the first one shows the distribution curves for the first—HAs isolated from fallow land soil during overgrowing, the second—HAs isolated from fallow land soils with different types of use, the third—HAs isolated from cropland soils, and the fourth—HAs isolated from indigenous soils.

The regions belonging to different fractions have different areas, which confirms the distribution of HAs in the three regions. The distribution curves have two distinct peaks characteristic of low and high MW regions. Among the MW distribution curves, the proportion of high MW fraction is 1.8–18.8% with Mr 301.1–432 kDa. The lowest content corresponds to HAs isolated from agricultural soil (Y10). The highest content of high MW fraction is observed in HAs isolated from native (Y3) and post-pyrogenic (Y4) soil; it may be associated with low maturity of the HAs and the quality of humification precursors. The HAs of fallow soils in the process of self-overgrowing were characterized by a relatively high content of high MW fraction 14.1–14.5% Mr 397.4–432 kDa. We can conclude that the transition of soils to fallow state leads to the transformation of the molecular structure of HAs towards HAs of native soils. At the same time, the HAs of fallow soils used in agriculture (as pasture and hayfield) (Y5–Y6) were characterized by a relatively low content of the high MW fraction of HAs 6.6–10.5% Mr 319–411 kDa, indicating that the reinvolvement of agriculture leads to an increase in new formation in the molecular structure of HAs and the maturity of organic matter. The obtained data are comparable to the data for arable tundra Cryosols of the European North of Russia, characterized by a higher contribution of the low MW fraction and a lower contribution of the high MW fraction compared to its natural analogue (Cryosols) [35]. Cryogenesis has a significant impact on the development of soils in the Far East; it is expressed in a decrease in the carbon content of HAs and an increase in the molecular mass of HAs, which is due to the low degree of humification of SOM [36].

As a result of the analysis of HAs from the soils of central Yakutia, it was revealed that in all studied preparations, the low MW fraction of HAs prevails (47.7–73.3%), which indicates the duration of the process of transformation of organic compounds as a result of humification and increasing age of organic molecules in the soil (Table 2). The increase in the time of organic molecules in soil is a characteristic stage of carbon dynamics in soil, which is due to the stabilization of organic matter. Thus, the most stable HA molecules are formed in the soils of arable land, namely, in the soils of the experimental vegetable garden and in the soils of arable land without signs of salinization. According to the polydispersity index of the HAs’ molecular composition, we also note that the highest values correspond to the territories of arable land (Y10), as well as the soil of the experimental school field (Y7). The polydispersity index is determined by a set of different molecular components by weight; respectively, the higher this index is, the more intensive the process of organic matter transformation. This is due to soil cultivation and the application of organic fertilizers.

The second place, by content of low MW fraction of Has, is occupied by fallow soils used as pastures (Y5). Thus, the use of fallow lands as pastures has a positive effect on the stability of organic matter. The HAs formed in fallow soils in the process of grassing and afforestation did not differ significantly; in the forest, the presence of molecules with relatively high MW is noted, which is associated with the presence in the organic matter of fatty acids, polysaccharides and polypeptides, which are the decay products of forest litter [34]. The lowest content of low MW fraction of HAs is characterized by native undisturbed soil, which indicates a rapid transformation process of organic residues in the soil. The same follows from the polydispersity index; here, it was the lowest among the studied HAs.

The distribution of the medium MW fraction of HAs differed slightly among the studied samples and was 21.4–34.8%; the lowest content was observed in the sample of HAs extracted from fallow soil under pasture, the highest content was observed in HAs of fallow soil under the hayfield. Such distribution indicates a high heterogeneity of conditions of formation of molecular composition of HAs in the studied soils. A relatively high content of low MW fraction of HAs in agricultural soils indicates relatively high rates of SOM transformation, which lead to the appearance of new formations in the composition of HAs, indicating the maturity of SOM. Principal component analysis (PCA) was performed to identify statistical dependence between the studied parameters (Figure 3).

According to the obtained graph, it was found that the content of the high MW fraction in the HA composition has a high correlation with the hydrogen content and increases depending on the increase in the H/C ratio. This may indicate that the high MW fraction of HA consists mainly of aliphatic structural fragments, which form the branched peripheral part of HA molecules. This is confirmed by Pearson’s correlation (H and high MW, r = 0.8; O and high MW, r = 0.92). At the same time, the low MW fraction of HAs depends on the carbon content (Pearson’s correlation C and low MW, r = 0.71), as well as on the polydispersity index; this is due to the wide range of molecular masses contained in HA molecules, which is caused by the degradation and the ability to form new molecular fragments in the composition of HA molecules [37].

Large-scale soil development and the establishment of agricultural land has led to increased rates of SOM mineralization and increased carbon dioxide emissions to the atmosphere [38]. The molecular structure of HAs allows us to assess the processes that occur as a result of SOM transformation and to estimate the rates of organic matter sequestration in soils [39]. It was found that HAs extracted from agricultural soils were characterized by the highest level of stabilization due to an increase in the content of the low molecular fraction, while natural soils were characterized by a relatively high content of the high molecular fraction, which is represented by alkyl compounds within HAs [40]. Our data are confirmed by Debska et al. [41], fertilization and soil cultivation leads to an increase in polydispersity index, which indicates the maturity of HA molecules, low values of polydispersity index in HA samples from native soils indicate an early stage of humification. However, scientists from Thailand have found that long-term and intensive use of agricultural soils has resulted in the loss of SOM due to land degradation, and one of the steps to restore fertility may be the transit of soils to fallow state [15]. Scientists who conducted studies on the qualitative composition of SOM found that the type of fertilizer plays a significant role in the formation of aliphatic structural fragments, as the use of manure leads to the formation of aliphatic periphery in the structure of HAs [11]. The relatively high content of the high MW fraction of HAs indicates the predominance of hydrophobic HA molecules, which prevent the dissolution and migration of organic compounds through the profile, thus protecting HAs from further transformation [42]. While the low-molecular fraction is characterized by the content of hydrophilic molecules that can migrate through the soil profile [15], in the extra-arid climate of central Yakutia, migration processes are less pronounced, so low-molecular compounds can accumulate in large quantities in the upper soil horizons. Long-term agriculture leads to a decrease in the MW of HA molecules and an increase in the content of low MW fragments. The age of soil transition to fallow state is not the leading factor determining the process of SOM stabilization and the formation of aromatic structural fragments in the composition of HAs; humification precursors and the duration of soil cultivation apparently play a more important role in the formation of stable molecular complexes [6].

## 3. Materials and Methods

### 3.1. Study Area

The soil sampling was carried out during fieldwork at the end of the summer of 2021. Soils were selected from arable, fallow as well as native lands. The area of study is shown in Figure 4.

Yakutsk is located in the Tuimaada valley on the left bank of the Lena River, in its middle stream. It is the largest city located in the permafrost zone. Soil sampling took place on the Prilensky plateau. It is composed of Cambrian and Ordovician gypsiferous and salinized limestones and dolomites [43,44]. The vegetation cover is represented by taiga pine and larch forests. The climate is sharply continental with long frosty, low-snow winters. The temperature drops to −45 °C in winter. Summers are moderately warm (15–17 °C), during which most precipitation falls. Precipitation is 350–450 mm per year [43,44].

Soil samples were taken on plots of fallow land in the process of self-overgrowing (Y1, Y2), native undisturbed (Y3, Y8), native, exposed to wildfire (Y4), fallow land under pasture (Y5), fallow land under hayfield (Y6), as well as arable land (Y7, Y9, Y10). In addition to sampling soils from fallow, natural and agricultural soils, soils formed in the first terrace of the river were considered, which is where the city’s agriculture is concentrated today. The description of the studied soil horizons is presented in Table 3.

The basic soil physicochemical characteristics are presented in Table 4. 

### 3.2. Sampling Strategy

Sampling was carried out on the territory of central Yakutia in the areas adjacent to fallow lands (Y1–Y2), reinvolved fallow lands (Y5–Y6), lands used in agriculture (Y7, Y9–Y10), and adjacent undisturbed lands (Y3, Y8), also soils were sampled in the territory exposed to wildfire (Y4) to assess the impact of fire on the qualitative composition of organic matter. Soils were sampled from the upper humus-accumulative horizons to analyze the composition of HAs. The level of hydromorphism of the territory is relatively low, the studied soils are formed on well-drained territories and there are no signs of overwatering.

### 3.3. Laboratory Analyses

All samples were dried at the +25 °C at the Department of Applied Ecology of SPSU and then sieved through 2 mm sieve with root removal. The pH values were determined by pH meter in H_2_O solution ratio 1:2.5. The carbon contents (C) were determined by Walkley–Black method [46]. The particle-size distribution analysis was carried out according to the Kachinsky “wet sedimentation” method, which is the Russian analogue of analysis by Bowman [47].

HAs were isolated from organo-mineral horizons of soils according to the method recommended by the International Society for the Study of Humic Substances (IHSS) with the modification of Vasilevich [33]. Decalcification was carried out with H_2_SO_4_ in a ratio of 1:10. The acid was neutralized with 1 M NaOH to pH = 7. The HAs were isolated from 25 g air-dried soil samples by 2-fold extraction with 0.1 M NaOH in ratio 1:10 for complete extraction of HAs. Then, HAs were precipitated with 1M H_2_SO_4_, adjusting the pH to 1.0. The HAs were purified from fulvic acids and other low MW compounds by dialysis and dried by heating at 35 °C in a forced convection laboratory oven. Deisolation was carried out using hydrofluoric acid.

The elemental composition of HAs was determined on a CHN analyzer (EA3028-HT EuroVector, Pravia PV, Italy). A van Krevelen diagram [37] was constructed from the elemental analysis data. Oxygen content was calculated from:(1)O=100−(C+H+N)
where C, H, N contents were obtained by CHN analyzer.

The degree of oxidation was calculated from:(2)$$w=2*(\left(O/16\right)-(H/1.01))/(C/12.01)$$

Analysis of the MW distribution of HAs was carried out in the Laboratory of Soil Chemistry of the Institute of Biology (Syktyvkar). The MW distributions of the HAs were obtained using an AKTAbasic 10 UPS chromatography system (Amersam Biosciences, Uppsala, Sweden) and a SuperdexTM 200 10/300 GL column (GE Healthcare Ltd., Danderyd, Sweden). The working range of the gel was determined using solutions of blue dextran and potassium bichromate at a concentration of 1 mg/cm^3^, and the column was calibrated using solutions of globular proteins. Preliminary solutions of HA preparations at a concentration of 0.1 mg/cm^3^ in 0.1 mol/dm^3^ NaOH were purified from low MW compounds by elution through a column of Sephadex G-10 gel. The eluate was Tris-HCl buffer, pH = 8.2, containing sodium dodecyl sulfate (0.1%) to prevent adsorption of HAs to the gel, NaCl (0.05 mol/dm^3^) to prevent superexclusion and maintain constant ionic strength, and sodium azide (0.02%) as an antibacterial agent. The original Unicorn 5.10 programme (GE Healthcare Ltd., Danderyd, Sweden) was used to process the chromatographic data obtained and to calculate the MW distribution of fractions of the HA preparations.

The following indices were used to analyze the MW distribution:

Number average *Mn*—averaging over the number of molecules in a biopolymer:(3)$$Mn=\frac{\sum n_{i}M_{i}}{\sum n_{i}},$$
where *n_i_* is the number of molecules with molecular mass *M_i_*.

Weight average *M_W_*—averaging over the mass of molecules in a biopolymer:(4)$$Mw=\frac{\sum n_{i}M_{i}^{2}}{\sum n_{i}M_{i}}$$
where *n_i_* is the number of molecules with molecular mass *M_i_*.

The average MW is calculated by the equation:(5)$$Mz=\frac{\sum n_{i}M_{i}^{3}}{\sum n_{i}M_{i}^{2}}$$
where *n_i_* is the number of molecules with molecular mass *M_i_*.

Polydispersity index—a quantitative characteristic of the degree of deviation of MW distribution from monodisperse. This index represents the ratio of weight average *M_W_* to number average *Mn*. 

## 4. Conclusions

The data on the molecular-mass distribution of humic acids in the soils of the fallow and agricultural lands of Central Yakutia developing under the action of cryogenesis were obtained for the first time. It was noted that soil transition to fallow state and the overgrowth of fallow lands lead to the transformation of the molecular composition of humic acids; there is a decrease in the content of low molecular weight fractions of humic acids, which indicates a low degree of transformation of organic molecules in the composition of the humic acids, and the stability of soil organic matter. It was found that the reinvolvement of the soils of fallow lands leads to an increase in the content of the low molecular weight fraction in the composition of humic acids, which indicates the formation of products of a “secondary” nature, which are more stable in the environment. The highest content of the low molecular weight fraction in the composition of humic acids was observed in agricultural soils; this is due to the active processes of the transformation of organic matter and the formation of molecular complexes resistant to biodegradation. The use of soils from fallow lands, which have not been affected by significant morphological and physicochemical changes in agriculture, can have a positive effect on the stabilization of soil organic matter and can lead to an increase in the maturity of soil organic matter.

## Figures and Tables

**Figure 1 molecules-29-03008-f001:**
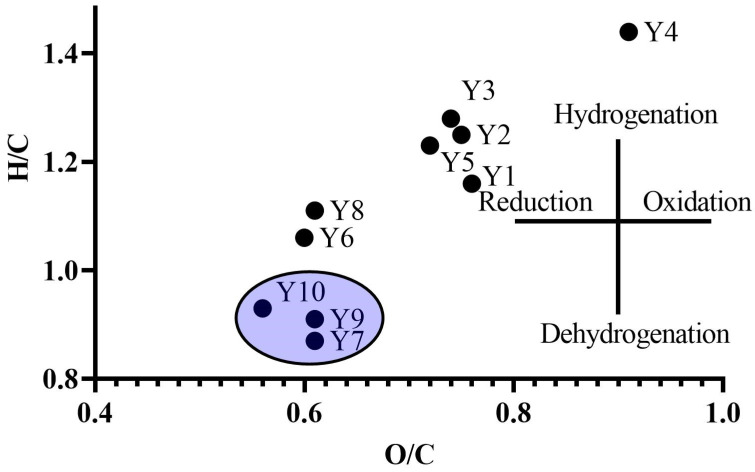
Van Krevelen diagram in H/C—O/C coordinates.

**Figure 2 molecules-29-03008-f002:**
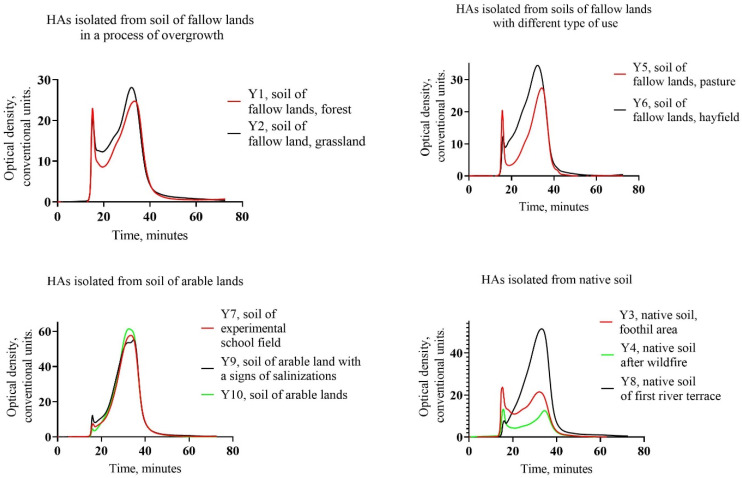
MW distribution of HAs isolated from soils of the central Yakutia.

**Figure 3 molecules-29-03008-f003:**
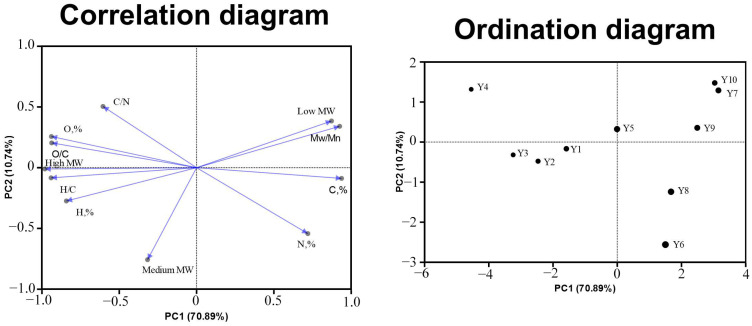
Results of principal component analysis for structural fragments and elemental composition of HAs.

**Figure 4 molecules-29-03008-f004:**
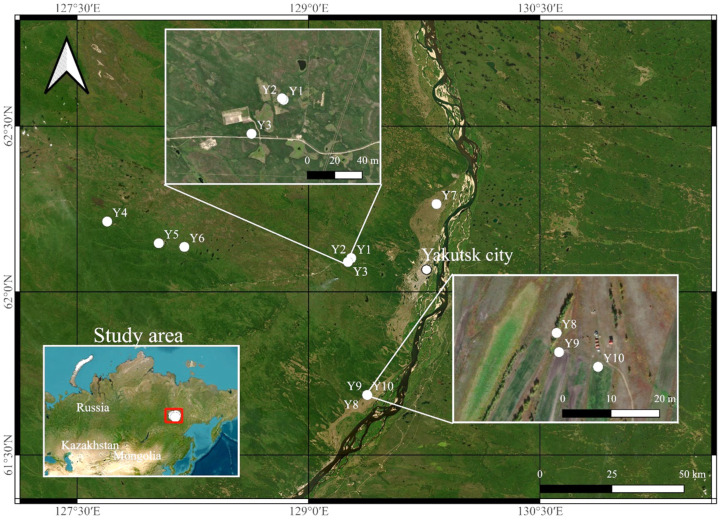
The study area of central Yakutia.

**Table 1 molecules-29-03008-t001:** Elemental composition of studied HAs isolated from soils of central Yakutia. The content of the investigated elements was recalculated on the deashed preparation. w Index—the degree of oxidation.

Soil ID	Mass Fraction, %				Molar Ratios			w
	N	C	H	O	C/N	H/C	O/C	
Y1	3.79 ± 0.01	45.57 ± 0.43	4.44 ± 0.05	46.20	14.03	1.16	0.76	0.36
Y2	3.27 ± 0.03	45.84 ± 0.28	4.81 ± 0.12	46.08	16.35	1.25	0.75	0.26
Y3	2.94 ± 0.03	46.36 ± 0.37	5.01 ± 0.05	45.69	18.42	1.28	0.74	0.20
Y4	2.78 ± 0.06	41.62 ± 1.11	5.03 ± 0.04	50.57	17.48	1.44	0.91	0.39
Y5	4.81 ± 0.02	46.16 ± 0.41	4.79 ± 0.03	44.24	11.19	1.23	0.72	0.21
Y6	5.62 ± 0.01	49.98 ± 0.17	4.45 ± 0.07	39.95	10.37	1.06	0.60	0.14
Y7	4.37 ± 0.04	50.61 ± 0.63	3.69 ± 0.24	41.33	13.51	0.87	0.61	0.36
Y8	5.05 ± 0.02	49.83 ± 0.04	4.66 ± 0.02	40.46	11.52	1.11	0.61	0.11
Y9	4.39 ± 0.02	50.69 ± 0.22	3.87 ± 0.04	41.05	13.46	0.91	0.61	0.31
Y10	3.81 ± 0.01	52.56 ± 0.11	4.09 ± 0.05	39.54	16.11	0.93	0.56	0.20

**Table 2 molecules-29-03008-t002:** The distribution of MW fractions in soils of the central Yakutia.

Soil ID	High Molecular Fraction				Medium Molecular Fraction				Low Molecular Fraction				*Mn*, kDa		*Mw*, kDa		*Mz*, kDa		*Mw*/*Mn*
	*Mr*, kDa	*S* *	Molecular Fraction, %		*Mr*, kDa	*S* *	Molecular Fraction, %		*Mr*, kDa	*S* *	Molecular Fraction, %								
			*x*	*S*			*x*	*S*			*x*	*S*	*x*	*S*	*x*	*S*	*x*	*S*	
У1	432	5	14.5	0.30	24.8	0.1	30.1	0.8	1.35	0.03	55.4	1.1	70.7	2.3	384	5	429	5	5.4
У2	397	0.9	14.1	0.06	28.3	0.2	34.0	0.5	1.58	0.03	51.9	0.5	66.4	0.4	339	0.7	393	1.0	5.1
У3	415	5	18.8	0.40	29.6	0.3	33.5	0.7	1.56	0.03	47.7	1.0	88.8	2.8	369	4	412	5	4.2
У4	409	4	18.6	0.80	29.3	1.0	26.8	0.6	1.07	0.02	54.6	1.5	84.0	3.0	371	4	406	4	4.4
У5	411	5	10.5	0.40	19.1	0.3	21.4	0.8	1.18	0.01	68.1	0.4	48.1	1.9	371	8	409	5	7.7
У6	319	2.2	6.6	0.10	24.3	0.1	34.8	0.9	1.55	0.04	58.6	0.8	30.5	0.2	228	5	310	2.7	7.5
У7	316	0.7	2.9	0.03	17.9	0.1	24.4	1.3	1.34	0.04	72.7	1.2	14.7	0.1	207	4	308	1.2	14.1
У8	311	0.7	3.6	0.05	21.3	0.1	30.9	0.6	1.40	0.01	65.6	0.5	18.6	0.1	194	2.8	300	1.1	10.4
У9	319	2.2	4.0	0.50	18.8	0.5	28	3	1.34	0.08	68	4	18.9	2.5	220	1.9	312	1.9	11.7
У10	301	0.1	1.8	0.06	18.0	0.1	24.9	1.6	1.39	0.05	73.3	1.6	10.9	0.5	156	2.0	287	0.6	14.4

* Standard deviation.

**Table 3 molecules-29-03008-t003:** Description of study soil horizon from central part of Yakutia.

Soil ID	Horizon	Depth, cm	Description	Location	Coordinates	Soil Name *
Y1	Abhp	6–30	Buried ploughing horizon with accumulation of humified organic matter, inclusion of coal	Fallow lands, beginning to overgrow with *Betula platyphylla*, *Populus tremuloides*	N 62.10211 E 129.2778	Plaggic Anthrosol (Loamic)
Y2	Abhp	0–20	Buried ploughing horizon with accumulation of humified organic matter, inclusion of coal	Fallow lands, overgrow by *Chamaenerion* sp.	N 62.10258 E 129.2766	Plaggic Anthrosol (Loamic)
Y3	Ah	4–15	Horizon with accumulation of poorly decomposed organic matter	Background forest with domination of *Larix dahurica*, *Betula platyphylla*	N 62.09103 E 129.2554	Calcic Cryosol (Loamic)
Y4	Ah	0–15	Pyrogenic material, dark, moist, sandy loam, humus leaks	Background forest with domination of *Larix dahurica*, *Betula platyphylla*	N 62.21319 E 127.6947	Calcic Cryosol (Loamic, Pyrogenic)
Y5	Ahp	6–10	Horizon with accumulation of humified organic matter, signs of overcompaction	Fallow lands, pasture	N 62.14756 E 128.029	Plaggic Anthrosol (Loamic)
Y6	Ahp	0–24	Ploughing horizon with accumulation of organic matter	Fallow lands, hayfield.	N 62.13653 E 128.1949	Plaggic Anthrosol (Loamic)
Y7	Ahp	0–26	Ploughing horizon with accumulation of humified organic matter	Experimental school field	N 62.26608 E 129.8299	Plaggic Anthrosol (Loamic)
Y8	Ah	0–6	Horizon with accumulation of humified organic matter	Background forest with domination of *Betula platyphylla*, *Chamaenerion* sp.	N 61.685317 E 129.379186	Calcic Cryosol (Loamic)
Y9	Ahp	0–15	Ploughing horizon with accumulation of humified organic matter, accumulation of salts on soil surface	Arable land without grasses	N 61.684961 E 129.379279	Plaggic Anthrosol (Loamic)
Y10	Ahp	0–25	Ploughing horizon with accumulation of humified organic matter	Arable land with oats.	N 61.68469 E 129.3808	Plaggic Anthrosol (Loamic)

* World reference base FAO [45].

**Table 4 molecules-29-03008-t004:** Physicochemical characteristics of studied soil.

Soil ID	Horizon	pH	C, %	Particle Size Distribution		
				Sand	Silt	Clay
Y1	Abhp	5.72	0.64	54	39	7
Y2	Abhp	5.69	1.19	55	43	2
Y3	Ah	5.23	6.51	72	26	2
Y4	Ah	5.15	2.53	81	14	5
Y5	Ahp	5.43	5.66	50	44	6
Y6	Ahp	5.47	3.85	47	47	6
Y7	Ahp	6.12	6.85	52	38	10
Y8	Ah	6.95	5.32	65	34	11
Y9	Ahp	6.47	6.86	60	35	5
Y10	Ahp	6.08	7.09	52	39	13

## Data Availability

The data of molecular weight distributions of HAs was obtained from the “Institute of Biology, Komi Science Center”. The data for CHN analysis was obtained from the “Center of Chemical Analyses and Materials”.

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
