# Peer review of "Molecular Weight Distribution of Humic Acids Isolated from Calcic Cryosol in Central Yakutia, Russia"

_molecules, 2024, doi:10.3390/molecules29133008_

Round 1

Reviewer 1 Report

Comments and Suggestions for Authors

1.     The structure of the manuscript was a bit messy. The order of results and methods was not correct.

2.     The backgrounds of the manuscript did not clearly describe the previous studies concerning composition of soil organic matter and the importance of molecular weights distribution of humic acids.

3.     The novelty of the manuscript was not clear, the current version was more like an investigation report instead of scientific paper. What the new in this manuscript should be emphasized.

4.     This study presented the work in central Yakutia, so far, it represented a regional work, which could not bring wider sights. To improve it, the significance of the work should be first described in the introduction and talked about more details in the results and discussion.

5.     Figure 2: what were the rule to categorize these 9 sites of soils into four figures.

6.     What were the correlations of soil characteristics among these 9 sites of soils?

7.     The reference should be updated with more newly published paper.

Author Response

Response to a review of the manuscriptMolecular weight distribution of humic acids isolated from Calcic Cryosol in Central Yakutia, Russia”.

Dear reviewer!

Thank you for your comments, they were completely taken into account, which improved the quality of the article for publication in Journal.

Text that has been changed is marked by yellow color.

General comments:

  1. The structure of the manuscript was a bit messy. The order of results and methods was not correct.

Response: Thank you the structure of article has been corrected.

  1. The backgrounds of the manuscript did not clearly describe the previous studies concerning composition of soil organic matter and the importance of molecular weights distribution of humic acids.

Response: Thank you! We improve the introduction section in aspect of the importance of investigation of humic acids.

  1. The novelty of the manuscript was not clear, the current version was more like an investigation report instead of scientific paper. What the new in this manuscript should be emphasized.

Response: Thank you! We improved the text of the article and added the novelty of work.

  1. This study presented the work in central Yakutia, so far, it represented a regional work, which could not bring wider sights. To improve it, the significance of the work should be first described in the introduction and talked about more details in the results and discussion.

Response: Thank you! We improved the Introduction section and added the global aspects of work. We have also expanded the Results and Discussion section.

  1. Figure 2: what were the rule to categorize these 9 sites of soils into four figures.

Response: Thank you! We added description in section “3.3 MW distribution in HAs isolated from soils of central Yakutia”.

  1. What were the correlations of soil characteristics among these 9 sites of soils?

Response: Thank you! In addition to PCA analysis, we confirmed the results by Pearson correlation.

  1. The reference should be updated with more newly published paper.

Response: Thank you! We updated the reference.

Thank you for work of our article.

Sincerely,

Professor of Saint-Petersburg State University, Evgeny V. Abakumov.

Junior researcher of Saint-Petersburg State University, Vyacheslav I. Polyakov.

Reviewer 2 Report

Comments and Suggestions for Authors

The manuscript investigated the molecular weight distribution of humic acids isolated from soils of central Yakutia, evaluated the degree of stabilization of SOM based on lthe elemental composition and MW distribution of HAs. There need some minor revision before publishing in the Journal.

1.Table 1 requires error values. And what w in the table means?

2.The results of the experiment are not very clear and should be supplemented. 

3.The author should add some new references, last three years.

Comments on the Quality of English Language

 Minor editing of English language required

Author Response

Response to a review of the manuscriptMolecular weight distribution of humic acids isolated from Calcic Cryosol in Central Yakutia, Russia”.

Dear reviewer!

Thank you for your comments, they were completely taken into account, which improved the quality of the article for publication in Journal.

Text that has been changed is marked by yellow color.

General comments:

  1. Table 1 requires error values. And what w in the table means?

Response: Thank you! We added the error values to Table 1. The “w” description has been added.

  1. The results of the experiment are not very clear and should be supplemented.

Response: Thank you! We changed the structure of the work, added additional information to Tables, improve introduction and result and discussion sections.

  1. The author should add some new references, last three years.

Response: The reference section has been updated.

Thank you for work of our article.

Sincerely,

Professor of Saint-Petersburg State University, Evgeny V. Abakumov.

Junior researcher of Saint-Petersburg State University, Vyacheslav I. Polyakov.

Reviewer 3 Report

Comments and Suggestions for Authors

The article is very important due to climate change. Unfortunately, it requires correction.

The article should be corrected - the order of the chapters is incorrect.

There are abbreviations in the introduction that are not explained. They can be used, but at the beginning you should provide the full name and the abbreviated form in brackets.

line 327 - can be deleted, explanation repeated in line 330

edit lines from 356 - 360. The explanations for the formulas Mw, Mn, Mz are the same

Conclusions - do not use abbreviations

Author Response

Response to a review of the manuscriptMolecular weight distribution of humic acids isolated from Calcic Cryosol in Central Yakutia, Russia”.

Dear reviewer!

Thank you for your comments, they were completely taken into account, which improved the quality of the article for publication in Journal.

Text that has been changed is marked by yellow color.

General comments:

  1. The article should be corrected - the order of the chapters is incorrect.

Response: Thant you! The chapters have been updated.

  1. There are abbreviations in the introduction that are not explained. They can be used, but at the beginning you should provide the full name and the abbreviated form in brackets.

Response: Thank you! We provide the explanation in Abstract section.

  1. line 327 - can be deleted, explanation repeated in line 330.

Response: Thank you! Done.

  1. edit lines from 356 - 360. The explanations for the formulas Mw, Mn, Mz are the same

Response: Thank you! Done.

5. Conclusions - do not use abbreviations

Response: Thank you! Done.

Thank you for work of our article.

Sincerely,

Professor of Saint-Petersburg State University, Evgeny V. Abakumov.

Junior researcher of Saint-Petersburg State University, Vyacheslav I. Polyakov.

Round 2

Reviewer 1 Report

Comments and Suggestions for Authors

The manuscript has been much improved.